# mpMRI-US Fusion-Guided Targeted Cryotherapy in Patients with Primary Localized Prostate Cancer: A Prospective Analysis of Oncological and Functional Outcomes

**DOI:** 10.3390/cancers14122988

**Published:** 2022-06-17

**Authors:** Esaú Fernández-Pascual, Celeste Manfredi, Cristina Martín, Claudio Martínez-Ballesteros, Carlos Balmori, Enrique Lledó-García, Luis Miguel Quintana, Raphael Curvo, Joaquín Carballido-Rodríguez, Fernando J. Bianco, Juan Ignacio Martínez-Salamanca

**Affiliations:** 1LYX Institute of Urology, Faculty of Medicine, Universidad Francisco de Vitoria, 28223 Madrid, Spain; esau.fernandez@salud.madrid.org (E.F.-P.); crismvivas@yahoo.es (C.M.); cmartinez@lyxurologia.com (C.M.-B.); balmoriboticario@gmail.com (C.B.); 2Department of Urology, Hospital Universitario La Paz, 28046 Madrid, Spain; 3Urology Unit, Department of Woman, Child and General and Specialized Surgery, University of Campania “Luigi Vanvitelli”, 80131 Naples, Italy; celeste.manfredi@unicampania.it; 4Department of Urology, Hospital Universitario Puerta De Hierro-Majadahonda, 28222 Madrid, Spain; raphael.deoliveira@salud.madrid.org (R.C.); jcarballido.hpth@salud.madrid.org (J.C.-R.); 5Department of Urology, Hospital General Universitario Gregorio Marañón, 28007 Madrid, Spain; enrique.lledo@salud.madrid.org; 6Department of Urology, Hospital Universitario Fundación Jiménez Díaz, 28040 Madrid, Spain; luis.quintana@quironsalud.es; 7Urological Research Network, Miami, FL 33016, USA; drbianco@research.surgery

**Keywords:** cryotherapy, focal therapy, predictor, prostate cancer, recurrence

## Abstract

**Simple Summary:**

Targeted cryotherapy is an emerging treatment for prostate cancer (PCa). mpMRI is a powerful tool for image fusion techniques that deliver incremental precision in diagnostic and treatment of PCa. Fusion targeted cryotherapy (FTC) arises from the simultaneous application of both these procedures. Recurrence is a concern after any type of PCa treatment, especially after targeted treatments. In this article we investigate the recurrence rate after FTC and the role of Prostate-Specific Antigen (PSA) as a predictor of recurrences. Our research provides new evidence on the feasibility of FCT by providing new insights on patient management.

**Abstract:**

Targeted therapy (TT) for prostate cancer (PCa) aims to ablate the malignant lesion with an adequate margin of safety in order to obtain similar oncological outcomes, but with less toxicity than radical treatments. The main aim of this study was to evaluate the recurrence rate (RR) in patients with primary localized PCa undergoing mpMRI/US fusion targeted cryotherapy (FTC). A secondary objective was to evaluate prostate-specific antigen (PSA) as a predictor of recurrences. We designed a prospective single-center single-cohort study. Patients with primary localized PCa, mono or multifocal lesions, PSA ≤ 15 ng/mL, and a Gleason score (GS) ≤ 4 + 3 undergoing FTC were enrolled. RR was chosen as the primary outcome. Recurrence was defined as the presence of clinically significant prostate cancer in the treated areas. PSA values measured at different times were tested as predictors of recurrence. Continuous variables were assessed with the Bayesian *t*-test and categorical assessments with the chix-squared test. Univariate and logistic regression assessment were used for predictions. A total of 75 cases were included in the study. Ten subjects developed a recurrence (RR: 15.2%), while fifty-six (84.8%) patients showed a recurrence-free status. A %PSA drop of 31.5% during the first 12 months after treatment predicted a recurrence with a sensitivity of 53.8% and a specificity of 79.2%. A PSA drop of 55.3% 12 months after treatment predicted a recurrence with a sensitivity of 91.7% and a specificity of 51.9%. FTC for primary localized PCa seems to be associated with a low but not negligible percentage of recurrences. Serum PSA levels may have a role indicating RR.

## 1. Introduction

Targeted therapy (TT) for prostate cancer (PCa) seeks to ablate an image-visible, biopsy-confirmed malignant lesion with an adequate safety margin [1]. The main purpose of TT is to limit toxicity in treating the entire prostate gland and rather focusing the tumor lysis on the areas of the prostate that harbors cancer cells. Therefore, TT spares neurovascular bundles, the sphincter, and the urethra [2]. TT is currently viewed as an emergent alternative to active surveillance or radical treatments (such as radical prostatectomy and radiotherapy) in selected patients with primary localized PCa [3]. Recently, TT has gained traction and is of most interest for the scientific community who have focused on safety and patient selection criteria; however, despite several consensus conferences, there are no standardized criteria supported by adequate scientific evidence [4]. According to current European Association Urology (EAU) guidelines, TT is still an investigational therapy; therefore, it should only be offered within clinical trials or well-designed prospective studies to subjects [5].

We share the perspective that the accurate detection of index lesion and tumor extension is essential for TT planning to avoid cancer persistence or recurrence. Likewise, these patients demand a rigorous follow-up protocol and must know they are at risk for persistent, recurrent or de novo disease [4,6]. Multiparametric magnetic resonance imaging (mpMRI) is considered the standard radiological modality for initial tumor identification, local staging, and for detection of disease recurrence [6,7]. Moreover, mpMRI has emerged as a powerful tool for image fusion techniques that deliver incremental precision in diagnostic and treatment of prostate cancer lesions. However, there is ample debate on the value of image-guided modalities [8,9].

Cryotherapy applied to the prostate has witnessed and passed the test of time, and currently represents a solid option for TT. Fusion targeted cryotherapy (FTC) induced apoptosis is achieved by the placement of cryo-needles in the target area through the perineum [10,11]. Cell death is the consequence of coagulative necrosis by protein denaturation, direct rupture of cellular membranes, and vascular stasis and microthrombi resulting in ischemic apoptosis [12].

The main aim of this study was to evaluate the recurrence rate (RR) in patients with primary localized PCa undergoing FTC. Secondary objectives were to evaluate prostate-specific antigen (PSA) as a predictor of recurrences and to describe adverse events (AEs), such as urinary and erectile function in the same patient cohort.

## 2. Materials and Methods

### 2.1. Study Design and Patient Enrollment

We designed a prospective single-arm study evaluating patients undergoing FTC. The study incorporated the Declaration of Helsinki on ethical principles for medical research involving human subjects, and all participants signed an informed consent for the inclusion in the study and the publication of data [13]. The Ethics Committee of our institution approved the research protocol in February 2013. Patient eligibility consisted of a diagnosis of primary localized PCa after a mpMRI/TRUS fusion-guided transperineal prostate biopsy that incorporated both systematic and targeted sampling, a Gleason score (GS) ≤ 4 + 3, with single or multifocal lesions, and a serum PSA ≤ 15 ng/mL. Stage cT2c was not an exclusion criterion. Exclusion criteria included any prior hormonal deprivation, chronic urinary retention, or lack of sexual activity. Patients underwent thorough council and all declined surgery or radiation treatments.

### 2.2. Patient Assessment and Measured Outcomes

All patients underwent medical history, digital rectal examination, PSA measurements, and mpMRI before prostate biopsy. mpMRIs were interpreted by a single radiologist initially using a likelihood system, evolving into the Prostate Imaging Reporting and Data System (PI-RADS) and adjusted accordingly overtime [14]. Prostate volume was determined by prostate segmentation of the mpMRI. An mpMRI/TRUS fusion-guided transperineal prostate biopsy was performed to diagnose PCa in all patients by experienced urologists combining systematic and targeted sampling. Index lesion was defined as the tumor focus with the highest GS detected with the biopsy; when two tumor foci had the same GS, the largest one was classified as the index lesion [15]. The main features of the index lesion were recorded for each subject.

After FTC, follow-up consisted of PSA (every 3 months the first year, every 6 months thereafter), mpMRI (every 6 months to a year, and thereafter according to PSA findings, usually annually), and fusion biopsy. However, biopsy was mandated if and when the mpMRI showed contrast enhancement in the treated area. Recurrence was defined as infield lesion with clinically significant PCa (GS ≥ 7). RR was chosen as the primary outcome. After 2019, we amended the protocol. Prostate biopsy was offered annually, but patients could avoid it if their serum PSA was stable (an increase of less than 2.0 ng/mL compared to the nadir [16]) or the mpMRI showed lack of enhancement on treated areas and no suspicion in untreated areas. We considered these cases as recurrence-free. PSA levels were evaluated at different times (baseline, at 12 months, after 12 months) to establish values that could be used as predictors of recurrence. The International Prostate Symptom Score (IPSS) [17], the Incontinence Questionnaire Short-Form (ICIQ-SF) [18], and Erectile Function-Erectile Function domain (IIEF-EF) [19] were recorded before cryotherapy and at 6 and 12 months. Versions of these questionnaires validated in the native language of patients were used. In addition, the need for phosphodiesterase type 5 inhibitors (PDE5i) and pads were noted. The patient continence was classified according to the number of pads used in 24 h as complete continence (0 pads), social continence (0–1 pad), or incontinence (≥2 pads) [20]. All postoperative AEs were reported according to the Clavien–Dindo (CD) classification [21]. Patients were followed-up until recurrence development, or at least 10 years after TT.

### 2.3. Surgical Procedure

All patients underwent FTC according to an individualized protocol based on prostate volume, number, size, and location of lesions. Patients were placed in a lithotomy position, underwent antibiotic prophylaxis with cephalosporins, and were anesthetized by deep sedation with spontaneous breathing. We utilized the stand-alone MIM Symphony™ treatment planning software (MIM Software Inc., Cleveland, OH, USA), EX3™ Stepper & Micro-Touch Stabilizer (CIVCO Medical Solutions, Coralville, IA, USA), Hitachi HI VISION Avius^®^ ultrasound machine (Hitachi, Tokyo, Japan) and the VISUAL ICE™ cryoablation system (Boston Scientific, Marlborough, MA, USA) to ablate the prostate target areas. Systematically, two-10 min/cycles (cooling–heating) on the target were applied. Thermocouples or urethral warmers were not uniformly used, but rather at the surgeon’s choice. We used real-time fusion for treatment monitoring. All procedures were performed by two experienced urologists in an outpatient setting.

### 2.4. Statistics

The categorical variables were reported as frequencies and percentages. The quantitative variables were described as means and standard deviations (SD), or medians and interquartile ranges (IQRs) in case of normal or non-normal distribution, respectively. The Kolmogorov–Smirnov test was used to assess the sample distribution [22]. Bayesian parametric statistics and chi-squared tests for non-parametric assessments were used for data analysis. Receiver operating characteristic (ROC) analysis was used to identify optimal prediction models lying on the ROC surface [23]. All tests were two-sided with a significance set at *p* < 0.05. Minimal clinically-important differences for ICIQ-SF, IPSS, and IIEF-EF were defined as ≥4, ≥3, and ≥4, respectively [24,25,26]. IBM SPSS Statistics (IBM Corp Released 2015. IBM SPSS Statistics for Windows, Version 23.0. Armonk, NY, USA: IBM Corp) was used for the statistical analyses.

## 3. Results

A total of 75 subjects were included in the study from February 2015 to April 2019 at our center (LYX Institute of Urology, Madrid, Spain). The baseline characteristics of patients are reported in Table 1. A single lesion was detected in 57 (76%) cases. Two and three lesions were treated in 16 (21.3%) and 2 (2.7%) patients, respectively. The main characteristics of index lesions are described in Table 2.

The median (IQR) time between prostate biopsy and FTC was 49 (35–65) days. The median (IQR) follow-up was 25 (18–38) months. Six (8%) patients were lost before 12 months of follow up for reasons unrelated to the study. Sixteen (23.2%) subjects refused a re-biopsy due to the low recurrence suspicion provided by mpMRI. Three (4.3%) patients were diagnosed with extraprostatic progression within the first year of follow-up. A total of 23/50 (46%) patients undergoing re-biopsy were positive for PCa. The lesions found with prostatic re-biopsies are detailed in Table 3. According to our definition, 10/66 subjects developed a recurrence (RR: 15.2%), while 56/66 (84.8%) patients showed a recurrence-free status. Recurrences were found infield or infield plus outfield in six and four subjects, respectively. Purely outfield lesions were not classified as recurrences, and were detected in 13 patients, although 4 of them were Gleason < 7. Median (IQR) time to recurrence was 24 (14–38) months.

Median (IQR) PSA after 3 months from FTC was 1.64 (0.42–4.01) ng/mL (75% decrease from baseline). PSA values measured during the first 12 months after treatment are reported in Figure 1. The median PSA at baseline was 6.56 ng/dL; it did not differ significantly (*p* = 0.290) between patients with and without recurrence. The median %PSA drop at one year was 65%. A statistically significant difference (*p* = 0.043) was found between median 12 months %PSA decline of subjects with and without recurrence, 30.2% and 68.6%, respectively. Likewise, the median PSA at recurrence was 4.1 ng/mL, and median PSA 12 months after FTC for those without recurrence was 2.7 ng/mL (*p* = 0.004). A %PSA drop of 31.5% during the first 12 months after treatment predicted a recurrence with a sensitivity of 53.8% and a specificity of 79.2% (Figure 2A). A PSA drop of 55.3% 12 months after treatment predicted a recurrence with a sensitivity of 91.7% and a specificity of 51.9% (Figure 2B).

At 12 months, we observed no significant differences (*p* > 0.05) in urinary functional outcomes when we compared to baseline in median IPSS and ICIQ-SF. We found a statistically but not clinically significant difference in IIEF-EF (*p* < 0.001), as 15/69 (21.7%) subjects had started taking or increased the dosage of PDE5i. Functional outcomes are summarized in Table 4. After 12 months, all 68 (100%) patients were continent.

No intraoperative complication was recorded. Urinary tract infections (CD grade II) occurred in five (7.2%) patients, being the most common AE. One (1.4%) subject experienced acute urinary retention (AUR) (CD grade IIIa) requiring bladder catheterization. Only two (2.9%) patients needed a surgical procedure for AEs (CD grade IIIa): one endoscopic intervention for lower urinary tract symptoms (LUTS) and one internal urethrotomy for urethral stricture.

## 4. Discussion

TT is emerging as an alternative to radical treatments for primary localized PCa in selected patients [10,27]. Its rationale is simple: avoid overtreatment and the associated toxicities that derive from such radical treatments. The fundamental questions for TT are whether its oncological outcomes are superior to surveillance and by how much and how far they would be or not from those of traditional therapeutic strategies. From a quality-of-life perspective, FTC is appealing due to an improved functional outcomes profile when compared to radical treatments [2]. Current evidence is promising but limited, and, therefore, the scientific community still considers TT an experimental approach that demands further studies determining efficacy and safety before its standardization in clinical practice [5]. However, the use of mpMRI and cognitive or software-based fusion imaging techniques for diagnosing PCa have become the standard. Their role as a tool for TT seems a probable consequence. This study, despite its limitations, suggests that TT is feasible and accomplishes the goal for 85% of patients, 1 year after treatment. The rationale is to treat proven prostatic cancer lesions and to spare healthy tissue. This is a familiar concept, one that is done routinely in the management of bladder cancer for example. TT can safely deliver oncological control with a favorable side effects profile, driven by less damage to surrounding healthy tissues [8,9].

Our robust series on FTC as a primary approach to localized PCa showed benefit at one year in 85% of the patients. Valerio et al. [28] evaluated the feasibility of FTC in 18 men with visible clinically significant PCa. Fourteen (77.8%) patients had primary PCa and four (22.2%) suffered from recurrence. Intermediate and high-risk cancer was detected in 13 (72.2%) and 5 (27.8%) patients, respectively. Late mpMRI showed no residual disease in the treated area (0% of infield lesions). In two patients, radiological progression of known contralateral disease was observed (11.1% of outfield lesions). Although the RR appears to be lower than in our study (0% vs. 15.2%), this could be explained by the failure to perform re-biopsies with a plausible underestimation of recurrences. Bergelson et al. [29] evaluated a cohort of 20 men undergoing FTC. mpMRI and re-biopsies after 6–9 months showed 77% of patients with an absence of clinically significant PCa. At 6 months, the authors described a median change in IIEF-5 and IPSS of −1.5 and 0 points, respectively. These results are consistent with ours. Like our study and that of Valerio et al. [28], no statistically significant difference (*p* = 0.12) between baseline and last follow-up at 12 months in IPSS scores were observed. Valerio et al. [28] described a non-significant incremental use of 17% in PDE5i by 12 months. Similar to our series, this increment is likely derived from the benefits of early use of these agents following radical prostate cancer treatments. A definitive answer demands randomized trials evaluating this critical functional outcome. Importantly, PSA significantly decreased (*p* = 0.001) 81.1%, from a baseline (9.54 ng/mL) to last follow-up at 12 months (1.8 ng/mL). While the role of PSA as a marker of success after TT is uncertain, our study sheds some light on what can be expected. We found no peer reviewed published studies assessing predictors of PCa recurrence after FTC. We found a published abstract correlating the use of PI-RADS 1 year after FTC with the likelihood of PCa [30]. However, as of this reporting the PI-RADS system is not intended for analysis of patients treated for PCa. Other investigators have evaluated the question of predictors after prostate hemiablation, Kongnyuy et al. [31] evaluated PSA as a predictor of biochemical recurrence (BCR) after primary hemiablation in 163 patients with localized PCa. BCR was defined using the Phoenix definition (PD) (PSA nadir + 2.0 ng/mL) and Stuttgart definition (SD) (PSA nadir + 1.2 ng/mL) [32]. According to PD and SD, BCR at a median time of 21.6 and 15.9 months occurred in 64 (39.5%) and 98 (60.5%) men, respectively. Survival analysis showed a 3-year BCR-free survival rate of 56% and 36% with PD and SD, respectively. Notwithstanding, neither the PD nor SD were modeled on partially treated prostates. Higher PSA nadir was an independent predictor of BCR with PD (*p* = 0.024), but not SD (*p* = 0.181). However, despite the differences between hemiablation and FTC, the data suggest a probable role for PSA discriminating good versus poor responders. The magnitude and what would be such a cut-off escapes us, and more research is warranted. It is also worth noting that we did not classify purely outfield lesions as recurrences; however, they were detected in 13 subjects and only 4/13 (30.8%) were not clinically significant. This highlights an inherent limitation of mpMRI and prostate biopsy rather than a low TT efficacy [8,9].

The main limitation is the single-arm design, which universally carries an intrinsic bias. Despite a prospective data collection, the series is small without a control arm. Other limitations to account for are the short follow-up, and the loss of several subjects during the study period may have prevented the recording of some events. The refusal of re-biopsy by a not negligible proportion of patients with a low risk of recurrence (stable PSA and unsuspected mpMRI) may have led to underestimating the RR. Failure to record all functional outcomes in some patients may have affected the reliability of the results. Prostate biopsy and re-biopsy are possible confounding factors that may have influenced some recorded outcomes. The arbitrary definitions of recurrence and recurrence-free status may limit the comparability of our results with those of other studies; however, to date, there are no standardized criteria to define recurrence in patients undergoing TT for PCa, and, therefore, arbitrary criteria are mandatory. Similarly, as the patient selection criteria and the follow-up protocol are not standardized, this does not facilitate comparison with other papers.

## 5. Conclusions

FTC for primary localized PCa seems associated with a low but not negligible percentage of recurrences. Moreover, it appears to be a safe procedure for postoperative AEs, urinary continence, urinary symptoms, and erectile function. PSA could be used as a predictor of recurrences after FTC by choosing cut-off values that allow for appropriate sensitivity and specificity according to the objectives set. Future large randomized clinical trials with long follow-ups are needed to confirm our findings. Furthermore, a particular effort should be directed to the definition of standardized criteria for the selection of patients, the diagnosis of recurrence, and the follow-up of the treated subjects to facilitate the comparability of future studies concerning FTC.

## Figures and Tables

**Figure 1 cancers-14-02988-f001:**
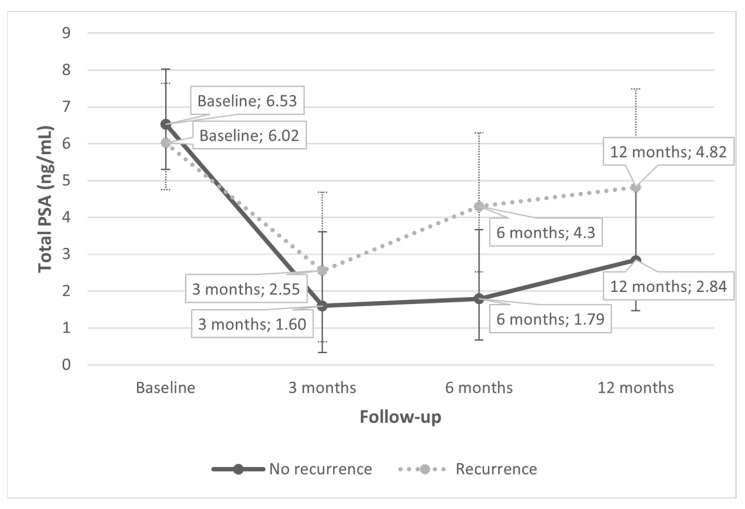
PSA during the first 12 months after FTC. PSA: Prostate-Specific Antigen; FTC: Fusion Targeted Cryotherapy. PSA values are reported as medians.

**Figure 2 cancers-14-02988-f002:**
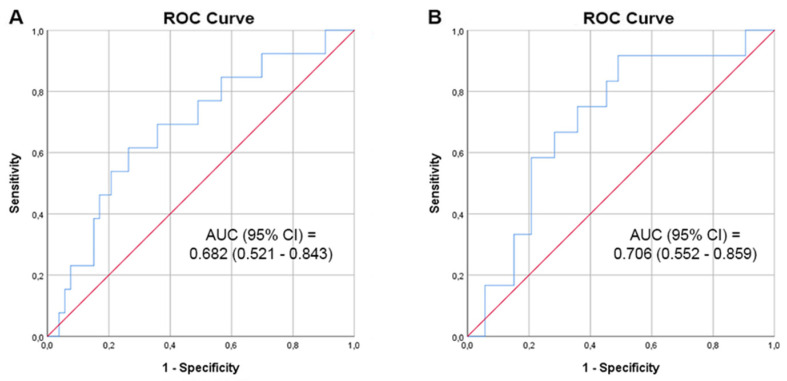
ROC curves for percentage of PSA reduction as predictor of PCa recurrence after FTC. Percentage of PSA reduction during the first 12 months (**A**) and after the first 12 months (**B**) ROC: Receiver Operating Characteristic; AUC: Area Under the Curve; CI: Confidence Interval; PSA: Prostate-Specific Antigen; PCa: Prostate Cancer; FTC: Fusion Targeted Cryotherapy.

**Table 1 cancers-14-02988-t001:** Baseline characteristics of patients.

Subjects, n	75
Age, years Median (IQR)	67 (62–72)
Prostate volume, cc Median (IQR)	50 (35–70.3)
Total PSA, ng/mL Median (IQR)	6.56 (5.1–8)
PSA-density, ng/mL/cc Median (IQR)	0.12 (0.08–0.20)
IPSS, points Median (IQR)	12 (7–16.8)
IIEF-EF, points Median (IQR)	17 (7–22)
ICIQ-SF, points Median (IQR)	0 (0–0)
Tumor stage, n (%)	T1c: 24 (32) T2a: 41 (54.7) T2b: 2 (2.7) T2c: 6 (8) NA: 2 (2.7)

IQR: InterQuartile Range; PSA: Prostate-Specific Antigen; IPSS: International Prostate Symptom Score (IPSS), IIEF-EF: International Index of Erectile Function-Erectile Function domain (IIEF-EF); ICIQ-SF: Incontinence Questionnaire Short-Form; NA: Not Available.

**Table 2 cancers-14-02988-t002:** Main characteristics of index lesions.

Volume, cc Median (IQR)	1.0 (0.54–2.47)
Size, mm Median (IQR)	12 (9–16.5)
Prostate segment, n (%)	Base: 12 (16) Middle: 44 (58.7) Apex: 18 (24) NA: 1 (1.3)
Prostate lobe, n (%)	Right lobe: 37 (49.3) Left lobe: 33 (44) Both lobes: 5 (6.7)
Prostate zone, n (%)	Peripheral: 57 (76) Transition: 12 (16) Anterior fibromuscular stroma: 5 (6.7) Central zone: 0 (0) NA: 1 (1.3)
PI-RADS score, n (%)	1: 0 (0) 2: 1 (1.3) 3: 11 (14.7) 4: 45 (60) 5: 17 (22.7) NA: 1 (1.3)
GS, n (%)	6: 33 (44) 7 (3 + 4): 23 (30.7) 7 (4 + 3): 19 (25.3)
EAU Risk Group, n (%)	Low: 19 (25.3) Intermediate: 48 (64) High: 6 (8) NA: 2 (2.7)

PI-RADS: Prostate Imaging Reporting and Data System; GS: Gleason Score; EAU: European Association of Urology; NA: Not Available.

**Table 3 cancers-14-02988-t003:** Lesions found with prostatic re-biopsies.

Infield lesions	GS 6: 2 GS 7 (3 + 4): 6 GS 7 (4 + 3): 2 GS 8: 2
Outfield lesions	GS 6: 7 GS 7 (3 + 4): 2 GS 7 (4 + 3): 5 GS 8: 2

GS: Gleason Score.

**Table 4 cancers-14-02988-t004:** Functional outcomes.

	Baseline	6 Months	12 Months	*p*-Value *
IPSS, points Median (IQR)	67 patients 12 (7–16.8)	67 patients 13 (9–16)	60 patients 12 (8–15)	0.176
IIEF-EF, points Median (IQR)	70 patients 17 (7–22)	70 patients 15 (7–19)	63 patients 16.5 (8–20)	<0.001
ICIQ-SF, points Median (IQR, min-max)	70 patients 0 (0–0, 0–4)	67 patients 0 (0–0, 0–6)	61 patients 0 (0–0, 0–5)	0.689

IPSS: International Prostate Symptom Score (IPSS), IIEF-EF: International Index of Erectile Function-Erectile Function domain (IIEF-EF); ICIQ-SF: Incontinence Questionnaire Short-Form. * Baseline (of patients still in follow-up at 12 months) vs. 12 months.

## Data Availability

Raw data are available upon specific request.

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
