# Peer review of "mpMRI-US Fusion-Guided Targeted Cryotherapy in Patients with Primary Localized Prostate Cancer: A Prospective Analysis of Oncological and Functional Outcomes"

_cancers, 2022, doi:10.3390/cancers14122988_

Round 1

Reviewer 1 Report

In this manuscript Fernandez-Pascual et al report the outcome of a single arm study on the effects of targeted cryotherapy for the treatment of primary localized Prostate Cancer (PCa). Since most PCa is localized identifying the best therapies would have far reaching consequences. Currently the two extreme approaches are prostatectomy or watchful waiting. The former is a radical approach with life changing negative consequences. The latter has the risk of having the cancer progress. This paper evaluated the middle of the road option of identifying the lesion and ablating it with cryotherapy. They followed PSA as a marker for re-occurrence and re-biopsied lesions.  The positive results are that this treatment had little negative impact on the patients. PSA was predictive at 1 year (statistically significant, but this was not very strong) so PSA measurements are of some value. However, since this was a single arm study it is difficult to ascertain the results (as the authors admit). One outstanding question is if the outcome is better than watchful waiting. The study cannot be redesigned, but the authors may have archival data on patients with matched characteristics, so they can use PSA and/or re biopsy data to compare results. This is not perfect but should provide some insight if this is of benefit. Intuitively, it would seem that ablating the largest lesion would be of benefit, even if it delays disease, so some data would improve this study and spur broader studies to establish this as an option.

Author Response

Dear Reviewers,

First of all, we wish to thank you for reviewing our paper entitled “mpMRI-US Fusion Guided Targeted Cryotherapy In Patients With Primary Localized Prostate Cancer: A Prospective Analysis Of Oncological And Functional Outcomes”. We realize that your time is valuable, and we are very grateful for your efforts.

We have carefully read all your comments and we have tried to clarify all the concerns expressed, improving the quality of the paper according to your suggestions. For your convenience, in addition to the revised manuscript, we report below our responses to your comments and the changes made in the manuscript.

Reviewer 1

Comment 1: One outstanding question is if the outcome is better than watchful waiting. The study cannot be redesigned, but the authors may have archival data on patients with matched characteristics, so they can use PSA and/or re biopsy data to compare results.

>>> Thank you for your comments. We assume that you are referring to active surveillance and not watchful waiting. We have not adequately collected and analyzed the patients managed with active surveillance in our series, but even so, they are not entirely comparable patients. In this study we have not treated with TT any Gleason 6 patient with very few affected cylinders and little affected length, in fact most of them are intermediate risk patients who could not be strict candidates for active surveillance, so we do not believe that the direct comparison of results between the two treatments can yield contradictory and inadequate results. Not to mention the fact that the objective of TT and active surveillance is not the same: with cryotherapy we seek to cure patients with minimal morbidity, while with active surveillance we aim to avoid overtreatment and morbidity of any kind to patients who do not need treatment of their low-risk tumor

Reviewer 2 Report

In the present manuscript, Fenandez-Pascual and colleagues report on a single institution study of 75 patients with localized prostate cancer who underwent targeted therapy with fusion targeted cryotherapy as their primary treatment.

This study represents an interesting approach for patients with localized prostate cancer. The authors contend that this may either be a viable approach patients for whom radiation therapy or radical prostatectomy is not felt to be suitable, or potentially in patients who would otherwise be undergoing active surveillance.

Several comments regarding the manuscript follow below.

Materials and Methods

Line 83-86:  The inclusion criteria for the trial should be clarified. It appears that the authors intended to enroll primarily patients with intermediate risk disease based on the ESMO prostate cancer guidelines.  However, they did not exclude cT2c disease, which is considered high risk, and which would typically be treated with radiation therapy and hormone therapy or radical prostatectomy. This difference should be clarified, as the statement in Lines 50-51, that TT may be an alternative to either active surveillance or radical treatments such as prostatectomy or radiation therapy suggests that TT may be applied in either of these populations. In table 2, six patients (8%) were classified as EAU high risk group disease and were offered TT. Can the authors address the potential concern for under treatment in the higher risk population?

Lines 100-103:  The frequency of mandated fusion biopsy is not clear from this paragraph, neither is the intended duration of follow-up. It appears that patients undergo multiparametric MRI either every 6 months or every year; the imaging schedule should be clarified. In addition, how often was repeat biopsy performed?

Line 104:  The primary endpoint of recurrence rate should be more precisely defined.  At what time point, or during what time frame was the recurrence rate measured? What was the overall duration of follow-up? Were patients only followed for a certain time period, or were all patients followed until they developed disease recurrence?

Results

In the provided manuscript, figures 1 and 2 are reported first, although tables 1 and 2 are referenced earlier.  It would be helpful to organize the tables and figures such that the graphics are represented in order of reference in the text.

Discussion

The conclusion is somewhat unclear – is the intended use of TT avoidance of other standard definitive therapies such as radiation therapy and radical prostatectomy, or as an alternative to surveillance? Further, the intended population for whom TT is felt to be most likely beneficial should be further defined.

Author Response

Dear Reviewers,

First of all, we wish to thank you for reviewing our paper entitled “mpMRI-US Fusion Guided Targeted Cryotherapy In Patients With Primary Localized Prostate Cancer: A Prospective Analysis Of Oncological And Functional Outcomes”. We realize that your time is valuable, and we are very grateful for your efforts.

We have carefully read all your comments and we have tried to clarify all the concerns expressed, improving the quality of the paper according to your suggestions. For your convenience, in addition to the revised manuscript, we report below our responses to your comments and the changes made in the manuscript.

Reviewer 2

Comment 1: Line 83-86:  The inclusion criteria for the trial should be clarified. It appears that the authors intended to enroll primarily patients with intermediate risk disease based on the ESMO prostate cancer guidelines.  However, they did not exclude cT2c disease, which is considered high risk, and which would typically be treated with radiation therapy and hormone therapy or radical prostatectomy. This difference should be clarified, as the statement in Lines 50-51, that TT may be an alternative to either active surveillance or radical treatments such as prostatectomy or radiation therapy suggests that TT may be applied in either of these populations. In table 2, six patients (8%) were classified as EAU high risk group disease and were offered TT. Can the authors address the potential concern for under treatment in the higher risk population?

>>> Thank you for appreciation. We have considered including patients with clinical stage cT2c who actually have palpable lesions but who on MRI are single lesions that exceed the midline or who present two palpable nodules. We believe that clinical staging with digital rectal examination should be revised and perhaps the general trend in the near future will be to rely more on MRI in pre-surgical staging, which offers much more information regarding the prognosis of the disease (involvement of the prostatic capsule, real size of the lesion, diagnosis of suspicious non-palpable lesions, etc.). For this reason, we do not believe that selected patients classified as "high risk" according to the EAU risk groups are actually of a worse prognosis than others who have, for example, a large lesion in a single lobe or two ipsilateral lesions (which would be classified as cT2b).

Comment 2: Lines 100-103:  The frequency of mandated fusion biopsy is not clear from this paragraph, neither is the intended duration of follow-up. It appears that patients undergo multiparametric MRI either every 6 months or every year; the imaging schedule should be clarified. In addition, how often was repeat biopsy performed?

>>> Thank you for the suggestion. mpMRI is performed every 6 months during the first year, and thereafter according to PSA findings, usually annually. The control prostate biopsy is offered one year after surgery and patients are encouraged to undergo rebiopsy despite having a favorable PSA dynamic and an MRI without a suspicious visible lesion. Once this negative control biopsy is performed, the criteria for rebiopsy is variable depending on PSA and MRI findings, although patients are offered to have it done annually. We have made some changes to make it clearer.

Comment 3: Line 104:  The primary endpoint of recurrence rate should be more precisely defined.  At what time point, or during what time frame was the recurrence rate measured? What was the overall duration of follow-up? Were patients only followed for a certain time period, or were all patients followed until they developed disease recurrence?

>>> Thank you for the comment. The recurrence rate is specified in the material and methods section and implies the anatomopathologic diagnosis of a Gleason greater or equal to 7 in a rebiopsy. This rebiopsy can be performed at any time after one year of treatment. In order to qualify a patient as "recurrence-free" we need a negative rebiopsy at one year or a suggestive PSA and MRI at one year of follow-up.

The median (IQR) follow-up was 25 (18-38) months. This data was already included in the text of the manuscript.

At the present time the follow-up does not have an end date, although it is likely to be maintained until at least 10 years without recurrence. The development of recurrence was cause for the discontinuation of this patient's follow-up. We replaced the term multivariate with multivariable in the text. We have made some changes to make it clearer.

Comment 4: In the provided manuscript, figures 1 and 2 are reported first, although tables 1 and 2 are referenced earlier.  It would be helpful to organize the tables and figures such that the graphics are represented in order of reference in the text.

>>> Thank you for the comment. The order in which the figures and tables appear is merely a matter of presentation of this manuscript, which, if the article is published, will be solved by inserting the figures and tables where appropriate in the text.

Comment 5: The conclusion is somewhat unclear – is the intended use of TT avoidance of other standard definitive therapies such as radiation therapy and radical prostatectomy, or as an alternative to surveillance? Further, the intended population for whom TT is felt to be most likely beneficial should be further defined.

>>> Thank you for the comment. As can be read in the conclusion, we do not believe that TT is a substitute treatment for any of the currently existing options, but just another option that should be used in appropriate cases to achieve the best results for the patient, both oncologically and functionally. The place it occupies in the treatments arsenal is right there, between active surveillance and radical treatment in localized PCa.

Sincerely,

The Authors

Round 2

Reviewer 1 Report

The authors  responded to and answered all of my concerns.